# Resource-efficient Inference with Foundation Model Programs

**Lunyiu Nie**[1]    **Zhimin Ding**[2]    **Kevin Yu**[1]    **Marco Cheung**[1]
**Christopher Jermaine**[2]    **Swarat Chaudhuri**[1]
[1]The University of Texas at Austin    [2]Rice University
lynie@utexas.edu, swarat@cs.utexas.edu

## Abstract

The inference-time resource costs of large language and vision models present a growing challenge in the deployment of agentic systems. We propose the use of *foundation model programs*, i.e., programs that can invoke foundation models with varying resource costs and performance, as an approach to this problem. Specifically, we present a method that translates a task into a program, then learns a policy for resource allocation that, on each input, selects foundation model "backends" for each program module. The policy uses smaller, cheaper backends to handle simpler subtasks, while allowing more complex subtasks to leverage larger, more capable models. We evaluate the method on two new "streaming" visual question-answering tasks in which a system answers a question on a sequence of inputs, receiving ground-truth feedback after each answer. Compared to monolithic multi-modal models, our implementation achieves up to 98% resource savings with minimal accuracy loss, demonstrating its potential for scalable and resource-efficient multi-modal inference [1].

## 1 Introduction

Foundation models (FMs) have reshaped the landscape of machine learning over the past few years, demonstrating unprecedented capabilities in language understanding (Achiam et al., 2023; Dubey et al., 2024), complex reasoning (Lu et al., 2024; Gupta et al., 2024), and multimodal tasks (Li et al., 2022a; Liu et al., 2024). While much of the community's attention has focused on their training costs, the *inference-time resource use* of FMs is increasingly becoming a practical bottleneck. For commercial applications that require real-time responses — for instance, continuous streams of user queries to a multimodal large language model (MLLM) — computational overhead and high latency can severely degrade user experience and inflate operational expenses (Xu et al., 2024).

In this paper, we propose the use of *foundation model programs* (FMPs) — code in Python-like languages that can call into a variety of specialized vision and language models as subroutines — as a way to address this problem. Such programs have been previously motivated by the interpretability and composability they bring to multi-step tasks (Surís et al., 2023; Gupta & Kembhavi, 2023; Subramanian et al., 2023) and agentic AI systems (Wu et al., 2023; Khattab et al., 2023). Our insight is that they can also enable fine-grained decisions about resource allocation: **simpler subtasks can rely on smaller, cheaper backends while more complex components can leverage larger, more capable models**.

Concretely, we propose a framework of *resource-efficient foundation model programming* in which a task is automatically translated into an FMP that captures subtask dependencies and conditional control flow. The framework is modality-agnostic, making it directly applicable to any domain — text, vision, speech, or multimodal — as long as the task can be expressed as an FMP. Each submodule of the program is then assigned to one of several backend models, differing in resource cost and capability. For example, consider the scenario in

---
[1]Source code and benchmarks are available at https://github.com/Flitternie/FMProgramming.

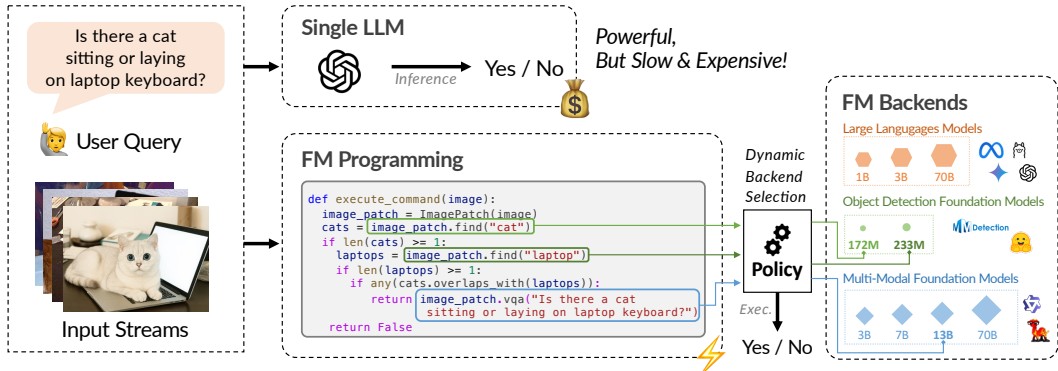

Figure 1: Illustration of an FM program synthesizing a VQA task by decomposing the task into sub-components. At runtime, the resource-efficient FM programming framework dynamically selects FM backends based on the task and input complexity to optimize accuracy and resource efficiency in real-time processing.

Figure 1, in which a visual question answering (VQA) system receives the query "*Is there a cat sitting or laying on a laptop keyboard?*" Here, our method generates a program that uses a small, inexpensive object detection model to check whether both a cat and a laptop are present. Only if that condition is met does it invoke a more powerful vision-language model (VLM) for finer-grained reasoning.

We specifically focus on "streaming" tasks in which the system repeatedly solves a task — for example, answering a question — on a *sequence* of inputs. An answer is generated without prior knowledge of the ground truth, then the system receives ground-truth feedback after each answer. In such settings, the cost of using a monolithic model is proportional to the number of inputs processed. By contrast, our approach uses the feedback from the early answers to learn a *policy* that dynamically selects which backend model to invoke for each subtask, conditioned on the program input. Specifically, we use a combination of a structured REINFORCE estimator and Thompson Sampling to learn this policy.

While existing routing or cascading strategies (Chen et al., 2023; Shnitzer et al., 2023; Lu et al., 2023; Nie et al., 2024) attempt to reduce large language model (LLM) inference overheads by switching between model sizes, they do not exploit the rich structural dependencies that arise in complex, compositional workflows. By contrast, our programs make these dependencies explicit, opening up opportunities for more flexible resource optimization.

Given the lack of standard benchmarks for resource-efficient sequential decision-making, we evaluate our approach on two newly introduced benchmarks: (1) a streaming binary VQA benchmark, where the questions require yes/no answers, spanning 33 compositional reasoning tasks with over 2,000 annotated images per task; and (2) a streaming open-form VQA benchmark, involving diverse questions with a broader answer space, covering 50 tasks with 500 annotated images per task. Experimental results show that our FMP-based system consistently reduces inference costs by 50% to 98% compared to one-size-fits-all baselines, without compromising task accuracy.

In summary, our contributions are as follows:

- We propose the use of *foundation model programs* as a flexible approach to cost-efficient inference for complex, multimodal workflows.
- We give a specific method for learning such programs in a sequential decision-making setting. The highlight of the method is an online resource allocation method that systematically trades off the resource consumption and performance of models in an input-dependent way.
- We release two streaming benchmarks for binary and open-form VQA, reflecting real-world tasks where inputs arrive sequentially at scale and resource-efficiency is key.
- We show empirical results on these benchmarks, which demonstrate that our program-based approach can achieve up to 98% cost savings with minimal accuracy degradation.

## 2 Problem Formulation

**Foundation Model Programs.**   We consider programs written in a language such as Python, potentially synthesized by an LLM. The programs are *neurosymbolic* because they interleave symbolic control flow with calls to a fixed set of *generic neural functions* $F = \{f_1, f_2, \ldots, f_K\}$, where each $f_k$ denotes a high-level functionality (*e.g.*, object detection, visual question answering, natural language understanding). For example, the program in Figure 1 makes calls to two generic neural functions `ImagePatch.find()` and `ImagePatch.vqa()`. In practice, we are interested in the case where the neural functions are implemented via foundation models. Hence, we use the term *foundation model program* (FMP) to refer to such programs.

Each generic function $f_k$ in an FMP has an associated set of $n_k$ *backend models* that can be used to implement it, namely $M_k = \{m_{k,1}, m_{k,2}, \ldots, m_{k,n_k}\}$, where each backend $m_{k,j}$ has different trade-offs between accuracy and computational cost. These backends may span a spectrum of models, from lightweight task-specific models to large, general-purpose language or multimodal models. Without loss of generality, we assume that the cost of invoking a backend is fixed and independent of the specific input it processes.

Now, assume that we analyze the program and produce an arbitrary ordering for the total of $N$ calls to generic neural functions, such that $k_i$ is the identity of the generic neural function associated with the $i$-th call in the program. Note that a program can call the same neural function multiple times with different inputs or arguments. The list of neural functions called by the program is then $\langle f_{k_1}, f_{k_2}, \ldots, f_{k_N} \rangle$. In our example program, the list is $\langle$`ImagePatch.find()`, `ImagePatch.find()`, `ImagePatch.vqa()`$\rangle$.

Further, assume the host programming language has a runtime in which we are able to dynamically assign each of the $N$ calls to generic neural functions to a particular backend. Let $j_i$ denote which of the backends is selected for the $i$-th generic neural function call, where $j_i \in \{1, \ldots, n_{k_i}\}$. That is, we select the $j$-th available neural backend $m_{k_i, j_i}$ for the $i$-th generic neural function call.

Thus, we can customize the behavior of an FMP to optimize for accuracy and runtime cost on a specific program input, by choosing a specific list of backends $\vec{v} = \langle m_{k_1, j_1}, \ldots, m_{k_N, j_N} \rangle$. We call $\vec{v}$ a *program configuration vector*. On input $x$, we use $p(x|\vec{v})$ to denote the output of the program, given that program configuration vector $\vec{v}$ was chosen.

**Generality.**   This framework generalizes the formulation of existing LLM resource management methods, including routing and cascading. Routing, as it is typically defined, is the problem of choosing a specific backend for a simple program that consists of a single generic neural function call: `lambda x : `$f$`(x)`. Cascading can be implemented as a foundation model program with nested `if-else` statements.

**Task objective.**   The idea of using FMPs for resource use optimization can be instantiated in a wide range of problems. In this paper, we focus on settings in which the goal is to solve a task — for example, answer a question — on a sequence of input-output pairs $\{x_t, y_t\}_{t=1}^T$. We assume that the structure of the programs we use only depends on the overall task and not the specific inputs. Therefore, on the input $x_t$ for time step $t$, we only need to decide on a suitable program configuration vector $\vec{v}_t$. We want $\vec{v}_t$ to be such that the program execution output $p(x_t \mid \vec{v}_t)$ approximates the ground truth $y_t$, while minimizing execution cost. To capture this trade-off, we define the following reward function:

$$R(\vec{v}_t, x_t, y_t) = -\mathcal{L}(p(x_t|\vec{v}_t), y_t) - \lambda \mathcal{C}(\vec{v}_t),$$

where $\mathcal{L}$ quantifies the output discrepancy between $p(x_t \mid \vec{v}_t)$ and the ground truth $y_t$, $\mathcal{C}$ represents the actual computational cost incurred when running the program with configuration $\vec{v}_t$ on input $x_t$, and $\lambda > 0$ is a trade-off weighting factor. Importantly, due to control flow in the program, not all neural backends specified in $\vec{v}_t$ may be invoked on a given input; $\mathcal{C}$ accounts only for the cost of the operations actually executed.

Over $T$ time steps, the objective is to learn a *policy* $\pi$ that maps each input $x_t$ to a program configuration vector $\vec{v}_t$. Let $\Pi$ be the space of such policies. We seek to solve the problem

$$\max_{\pi \in \Pi} \sum_{t=1}^{T} R(\vec{v}_t, x_t, y_t) \quad \text{subject to} \quad \vec{v}_t = \pi(x_t). \tag{1}$$

Importantly, we require this optimization problem to be solved *online*. That is, we assume that our inputs arrive sequentially and require decisions to be made without prior knowledge of the ground truth. Only after the selected configuration is executed and the output $p(x_t \mid \vec{v}_t)$ is produced is the ground truth $y_t$ revealed and the reward $R(\vec{v}_t, x_t, y_t)$ computed.

## 3 Methodology

Our approach to addressing this problem consists of two key phases: *offline code generation* and *online resource allocation*.

### 3.1 Offline Code Generation

We begin by synthesizing a foundation model program $p$ from the user specification using an LLM. This process yields a task-specific program sketch, including a sequence of generic neural function calls $\{f_{k_1}, f_{k_2}, \ldots, f_{k_N}\}$, which defines the high-level structure of the computation, while the backend selection for these neural functions is determined online.

The correctness of the FMP is critical, as it defines the entire computational workflow for the task. To ensure transparency and verifiability, FMPs are constructed with an explicit neurosymbolic representation, making them straightforward to inspect, validate, and debug. Although we employ an LLM to generate these programs, our framework is designed to be agnostic to the source of the FMP and fully supports human-authored workflows.

### 3.2 Online Resource Allocation

After offline synthesis, the main challenge is to select a program configuration vector $\vec{v}_t$ for each input $x_t$, dynamically assigning a neural backend $m_{k_i, j_i} \in M_{k_i}$ to each function call $f_{k_i}$. The space of possible configurations grows combinatorially with $N$, making exhaustive search intractable. To address this, we propose a structured policy that decomposes this decision process into $N$ manageable sub-policies, one per function call.

Specifically, for each function call $f_{k_i}$, we define a *sub-reward* function:

$$r_{k_i, j_i} = -\lambda \mathcal{C}(m_{k_i, j_i}) - \frac{1}{N} \mathcal{L}(p(x_t | \vec{v}_t), y_t) \quad \text{where} \quad m_{k_i, j_i} \in \vec{v}_t.$$

This sub-reward decomposes the global reward $R(\vec{v}_t, x_t, y_t)$ into structured contributions. It integrates the local computational cost $\mathcal{C}(m_{k_i, j_i})$ associated with backend $m_{k_i, j_i}$, and a portion of the predictive loss $\mathcal{L}(p(x_t | \vec{v}_t), y_t)$, that is determined when the program $p$ is executed with the entire configuration $\vec{v}_t$.

To model these rewards, we define a *subpolicy* $\pi_{k_i}$ with learnable parameters $\theta_{k_i}$. Given the input $x_t$, subpolicy $\pi_{k_i}$ outputs a reward prediction

$$r'_{k_i, j_i} = \pi_{k_i}(m_{k_i, j_i} | x_t; \theta_{k_i}) \quad \text{for each } m_{k_i, j_i} \in M_{k_i}.$$

This structured design simplifies the decision space: rather than jointly optimizing over all $n_{k_1} \times \cdots \times n_{k_N}$ backend combinations, we train $N$ separate subpolicies. Each subpolicy is specialized to one of the $N$ function calls, thereby simplifying the optimization process, enabling parallel learning, and reducing unwanted interference across calls. This per-call decomposition is critical for a precise credit assignment. In a monolithic joint policy, it is difficult to attribute rewards correctly, especially in the presence of control flow where some function calls may be conditionally skipped. Our structured approach resolves this

---

**Algorithm 1** Structured REINFORCE Framework

---

**Initialize:** Policy parameters $\theta_{k_i}$, uncertainty estimates $\mathbf{U}_{k_i}$, learning rate $\eta$, exploration factor $\nu$
**for** each input $x_t$ **do**
    **for** each function call $f_{k_i}$ in program $p$ **do**
        Predict reward $r'_{k_i,j_i} = \pi_{k_i}(m_{k_i,j_i}|x_t; \theta_{k_i})$ for each FM backend $m_{k_i,j_i}$
        Compute uncertainty $\sigma_{k_i,j_i} = \sqrt{\sum_l \frac{g^2_{k_i,j_i,l}}{\mathbf{U}_{k_i,l}}}$
        Sample adjusted reward for exploration:   $\hat{r}_{k_i,j_i} \sim \mathcal{N}\left(r'_{k_i,j_i}, (\nu \cdot \sigma_{k_i,j_i})^2\right)$
        Select FM backend $m_{k_i,j_i^*}$ with highest sampled reward $\hat{r}_{k_i,j_i}$
        Update parameter uncertainties $\mathbf{U}_{k_i,l}$
    **end for**
    Execute program $p$ with selected configuration $\vec{v}_t = (m_{k_1,j_1^*}, ..., m_{k_N,j_N^*})$
    Observe final reward $R(\vec{v}_t, x_t, y_t)$
    **for** each function call $f_{k_i}$ **do**
        Compute policy gradient $\nabla_{\theta_{k_i}} \mathcal{J}(\pi_{k_i})$ based on observed reward
        Update policy parameters:   $\theta_{k_i} \leftarrow \theta_{k_i} - \eta \nabla_{\theta_{k_i}} \mathcal{J}(\pi_{k_i})$
    **end for**
**end for**

---

by updating a subpolicy only when its corresponding function is executed, ensuring that reward signals are correctly aligned with the decisions that caused them.

**Gradient-based Thompson Sampling.** To balance exploration and exploitation in the online setting, decisions are made using Thompson Sampling (Zhang et al., 2020) instead of greedily selecting the FM backend with the highest predicted reward. For each backend $m_{k_i,j_i} \in M_{k_i}$, the subpolicy samples a reward from a normal distribution:

$$\hat{r}_{k_i,j_i} \sim \mathcal{N}\left(r'_{k_i,j_i}, (\nu \cdot \sigma_{k_i,j_i})^2\right),$$

where $\sigma_{k_i,j_i} = \sqrt{\sum_l \frac{g^2_{k_i,j_i,l}}{\mathbf{U}_{k_i,l}}}$ quantifies uncertainty in the reward prediction. Here, $l$ indexes each individual parameter of the subpolicy, $g_{k_i,j_i,l}$ is the gradient of the reward prediction with respect to parameter $l$, $\mathbf{U}_{k_i}$ tracks the accumulated gradient-based parameter uncertainties, and $\nu$ scales the exploration. We select the backend with the highest sampled reward:

$$j_i^* = \underset{j_i \in \{1, ..., n_{k_i}\}}{\arg\max} \hat{r}_{k_i,j_i}.$$

After selection, the uncertainty parameter $\mathbf{U}_{k_i}$ is updated to refine future exploration:

$$\mathbf{U}_{k_i,l} \leftarrow \mathbf{U}_{k_i,l} + g^2_{k_i,j_i^*,l},$$

where $g_{k_i,j_i^*,l}$ is the gradient of the selected backend $m_{k_i,j_i^*}$ with respect to parameter $l$.

We repeat the above process for each $i = 1, \ldots, N$, allowing each subpolicy to choose one backend per function call, thereby yielding the program configuration vector at time step $t$:

$$\vec{v}_t = (m_{k_1,j_1^*}, m_{k_2,j_2^*}, \ldots, m_{k_N,j_N^*}).$$

**Structured REINFORCE Algorithm.** We now describe the online learning of the subpolicies using only the global reward $R(\vec{v}_t, x_t, y_t)$ observed after execution. The learning objective for the overall policy $\pi = \{\pi_{k_1}, \ldots, \pi_{k_N}\}$ is to maximize cumulative reward over $T$ episodes:

$$\mathcal{J}(\pi) = \sum_{t=1}^{T} R(\vec{v}_t, x_t, y_t), \quad \text{where } \vec{v}_t = (m_{k_1,j_1^*}, m_{k_2,j_2^*}, \ldots, m_{k_N,j_N^*}).$$

Since the reward $R(\vec{v}_t, x_t, y_t)$ is equivalent to the aggregation of all sub-rewards:

$$R(\vec{v}_t, x_t, y_t) = -\mathcal{L}\big(p(x_t|\vec{v}_t),\, y_t\big)\, -\, \lambda\, \mathcal{C}\big(\vec{v}_t\big)$$

$$= \sum_{i=1}^{N}\Big[-\frac{1}{N}\mathcal{L}(p(x_t|\vec{v}_t), y_t) - \lambda\mathcal{C}(m_{k_i, j_i})\Big] = \sum_{i=1}^{N} r_{k_i, j_i^*},$$

we convert the learning objective into optimizing each subpolicy $\pi_{k_i}$ independently:

$$\mathcal{J}\big(\pi_{k_i}\big)\; =\; \sum_{t=1}^{T}\mathbb{E}_{j_i^* \sim \pi_{k_i}(\cdot|x_t)}\Big[r_{k_i, j_i^*}\Big].$$

Because the program execution is non-differentiable due to control flow structures like conditional branches and loops (Kreikemeyer & Andelfinger, 2023), we employ the REINFORCE algorithm (Williams, 1992) to estimate policy gradients:

$$\nabla_{\theta_{k_i}}\mathcal{J}\big(\pi_{k_i}\big)\; =\; \sum_{t=1}^{T}\mathbb{E}_{j_i^* \sim \pi_{k_i}(\cdot|x_t)}\Big[\nabla_{\theta_{k_i}} \log \pi_{k_i}\big(m_{k_i, j_i^*} \mid x_t; \theta_{k_i}\big)\, \cdot\, r_{k_i, j_i^*}\Big].$$

Note that each $r_{k_i, j_i^*}$ depends on the full program execution but reflects a partial credit assignment for subpolicy $\pi_{k_i}$.

In practice, we approximate the expectation using $S$ sampled trajectories:

$$\nabla_{\theta_{k_i}}\mathcal{J}\big(\pi_{k_i}\big)\; \approx\; \sum_{t=1}^{T}\sum_{s=1}^{S}\nabla_{\theta_{k_i}} \log \pi_{k_i}\big(m_{k_i, j_i^*}^{(s)} \mid x_t^{(s)}; \theta_{k_i}\big)\, \cdot\, r_{k_i, j_i^*}^{(s)}.$$

The sub-policies are periodically trained to stabilize learning:

$$\theta_{k_i} \leftarrow \theta_{k_i} - \eta\nabla_{\theta_{k_i}}\mathcal{J}(\pi_{k_i}),$$

where $\eta$ is the learning rate.

The overall framework is detailed in Algorithm 1 with a walk-through example in Appendix Figure 5. This methodology provides a tractable solution to the online resource allocation challenge, with a theoretical no-regret guarantee provided in Appendix A. By separating offline synthesis from online optimization and employing a decomposed policy structure, we achieve both flexibility and efficiency in program execution.

## 4 Benchmark

Motivated by the need for structured, sequential evaluation beyond the single-image-per-query setups typical of existing visual question answering (VQA) datasets (Goyal et al., 2017), we introduce two novel streaming VQA benchmarks. These benchmarks are designed to reflect real-world latency-critical deployments by simulating input streams, long-tail difficulty, class imbalance, and adversarial distractors. While partially synthetic, they are constructed using real data (e.g., COCO, GQA) and validated through LLM and human annotations to ensure complexity and robustness.

Our first **Streaming Binary VQA** benchmark focuses on yes/no question answering, a task commonly studied in previous works (Antol et al., 2015; Zhang et al., 2016; Hudson & Manning, 2019). In this benchmark, systems are challenged to determine whether a sequence of images satisfies complex, compositional queries. These queries incorporate diverse reasoning types—spatial (*e.g.*, "Is there a person riding a bicycle next to a bus on the street?"), logical (*e.g.*, "Are there people riding bikes, scooters, or motorcycles while holding or using umbrellas?"), and numerical (*e.g.*, "Are there at least four horses on a beach?")—to better reflect real-world reasoning demands. The final benchmark includes 33 queries with more than 2000 annotated images for each query, featuring a realistic class imbalance setup. Further details on the benchmark construction and evaluation are provided in Appendix C.1.

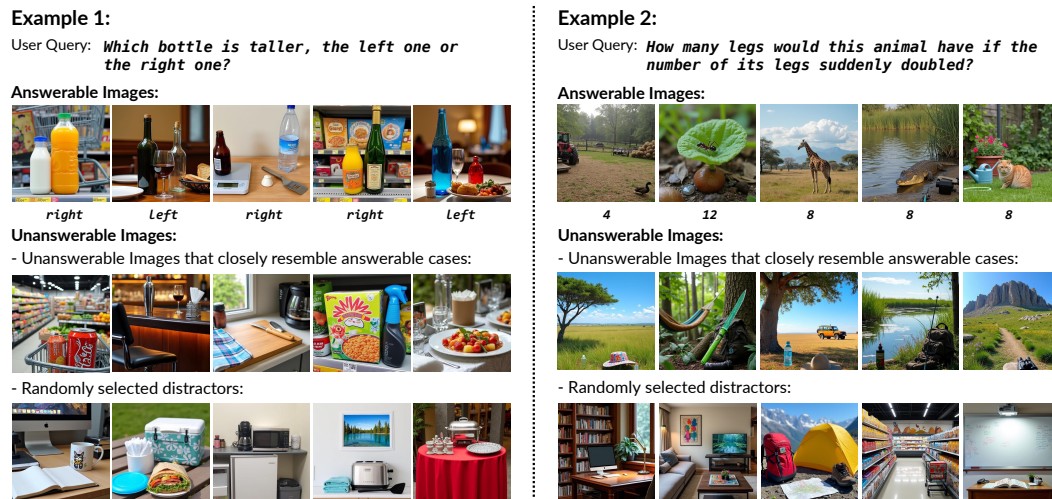

Figure 2: Examples of the synthetic images in the Streaming Open-form VQA benchmark.

Our second benchmark, **Streaming Open-form VQA**, evaluates a system's ability to answer open-form questions for a sequence of input images. This benchmark spans five reasoning categories: spatial (*e.g.*, "What is in the jar to the left of the juice?"), logical (*e.g.*, "What is the black object on the desk that is not electronic?"), numerical (*e.g.*, "How many extra bottles of beer do we need to make it a half dozen?"), comparative (*e.g.*, "Which bottle is taller, the left one or the right one?"), and external knowledge reasoning (*e.g.*, "How many states are there in the country whose flag is shown?"). Images are generated using a diffusion model with a dedicated pipeline to ensure diversity and quality control. To evaluate model robustness, we also introduce unanswerable images that are visually similar to the query but semantically invalid for answering. The final benchmark includes 50 queries with 500 annotated images per query. Complete details of the image generation pipeline, neurosymbolic program synthesis, and evaluation metrics (exact match accuracy) are described in Appendix C.2.

## 5 Experiments

### 5.1 Implementation Details and Experimental Setups

We implement the structured policy using ResNet-18 with ∼11M parameters (He et al., 2016), ensuring the policy training and inference overheads do not exceed the resource savings. During evaluation, our experiments use varying sizes of GLIP (Li et al., 2022b) and GroundingDINO (Liu et al., 2023) for object detection, OFA (Wang et al., 2022), BLIP-2 (Li et al., 2023), and Qwen2.5-VL series (Bai et al., 2025) for visual-language understanding, and Llama 3 series (Dubey et al., 2024) as the LLMs. We implement an instrumentation system that analyzes the programs to determine function calls, modifies the abstract syntax tree (AST) to inject program configurations, and executes the modified program while collecting performance metrics including the execution traces and the computational costs.

### 5.2 Baselines

We compare our system against several baselines to establish its effectiveness:

**Single MLLMs.** We evaluate our system against the state-of-the-art multimodal LLMs (MLLMs) that integrate both vision and language reasoning capabilities (Bai et al., 2025).

**MLLM Routing.** As an alternative adaptive strategy, a multi-armed bandit dynamically routes user queries to multimodal LLMs of varying sizes with different cost-accuracy trade-offs based on estimated rewards, balancing exploration and exploitation (Nguyen et al., 2024; Li, 2025). However, it does not account for the task structures in user queries.

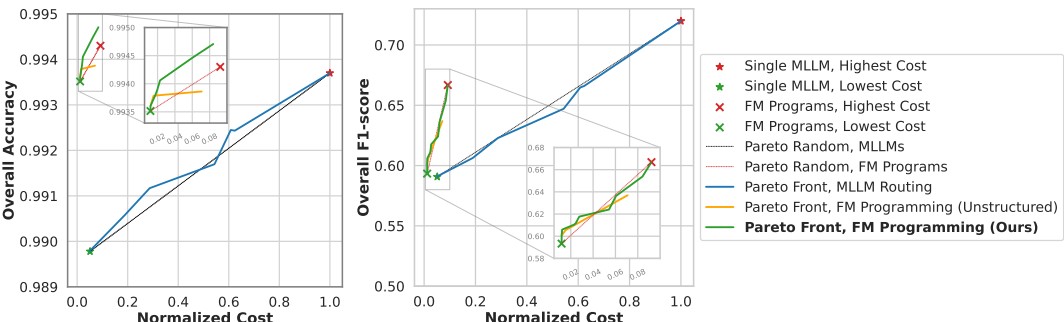

Figure 3: Experimental results on the Streaming Binary VQA benchmark. Costs are normalized based on the inference costs of the most expensive MLLM, *i.e.*, Qwen2.5-VL 72B.

**Static FM Program Configurations.** A common approach to resource management is to use a fixed, pre-determined configuration of foundation models for the FM programs without dynamic backend selection. Given the combinatorial space of configurations, we implement two variants: one using the cheapest FM configurations and another using the most expensive configurations.

**Pareto-Random Routing.** Following the standard practice in prior works (Hu et al., 2024; Jitkrittum et al., 2025), we employ a straightforward yet effective Pareto-random routing strategy through linear interpolation. We implement this approach separately for two scenarios: multimodal LLMs and static FM program configurations.

In the experiments reported in the main paper, we consistently use Qwen2.5-VL (3B and 72B) for both the Single MLLM and LLM Routing baselines. The FM program backends consist of Grounding-DINO Tiny (172M) and Base (224M) for object detection, along with Qwen2.5-VL 3B and 72B for vision-language understanding.

## 5.3 Results

**Streaming Binary VQA.** As shown in Figure 3, the FM Programming approach achieves accuracy comparable to the largest MLLM, while reducing computational costs by over 98%. This striking efficiency gain stems from its ability to exploit task structure: FM Programs use lightweight object detection modules to filter out most negative samples, drastically reducing expensive MLLM inference.

Beyond cost savings, FM Programming consistently delivers superior cost-accuracy and cost-F1 trade-offs, particularly in the low-cost regime. The Pareto frontier of FM Programming (green line) significantly outperforms both the Pareto Random baseline for FM Programs (red dashed line) and the LLM routing baseline (blue line), illustrating the effectiveness of structured decision-making via the Structured REINFORCE algorithm.

Compared to the LLM routing baseline, FM Programming achieves a 6% improvement in F1 score at similar computational cost (10% of the largest MLLM). While its maximum F1 score is lower than that of the Single MLLM and LLM routing baselines, this is due to conservative thresholds in the object detection modules that prioritize precision over recall to minimize costly false positives. As shown in Appendix Figure 8, the learned policy increasingly favors precision over recall as the cost budget grows. In practice, this trade-off can be adjusted by tuning the reward function used in Structured REINFORCE.

To evaluate robustness, we conduct additional experiments using an alternative set of FM backends. As shown in Appendix Figure 9, despite reduced backend capabilities leading to lower raw accuracy and F1 scores, FM Programming still maintains clear Pareto dominance over the baseline. This demonstrates its strong generalizability and adaptability across different FM backend configurations.

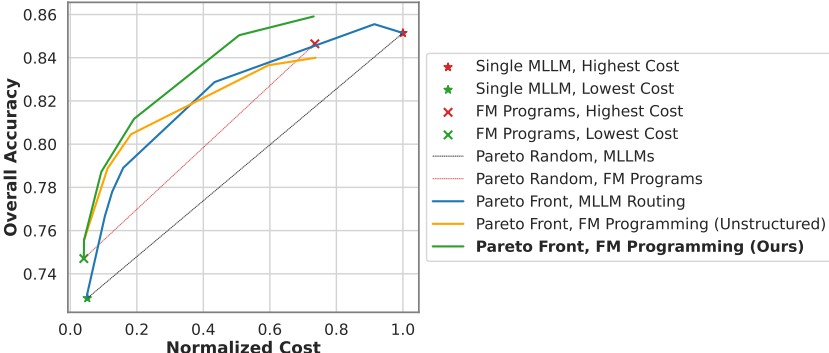

Figure 4: Experimental results on the Streaming Open-form VQA benchmark. Costs are normalized based on the inference costs of the most expensive MLLM, *i.e.*, Qwen2.5-VL 72B.

**Streaming Open-form VQA.** We further evaluate our method on the more challenging Streaming Open-form VQA benchmark. As shown in Figure 4, FM Programs (red dashed line) consistently outperform MLLMs (black dashed line) by over 2% in accuracy at equivalent cost, showcasing the advantage of modular programs over end-to-end models in leveraging task structures.

Building on this, the FM Programming approach (purple line) effectively expands the Pareto frontier. Through dynamic backend allocation with Structured REINFORCE, it achieves up to 50% cost savings without sacrificing performance compared to the largest MLLM, and in some cases even outperforms the MLLM while reducing cost by 30%.

Notably, FM Programming offers high marginal returns in low-cost regimes — small increases in cost yield substantial accuracy gains. It clearly surpasses the Pareto Random baseline and consistently outperforms the LLM Routing method, providing better cost-performance trade-offs. This demonstrates that the proposed FM Programming framework is effective in fully leveraging the task and input structure for optimal cost-efficiency.

**Ablation Study.** To assess our structured policy design, we compare it against an unstructured policy variant that jointly optimizes all backend selections. As shown in Figures 3 and 4, the unstructured policy (yellow line) significantly underperforms on both benchmarks, due to its inability to manage conditional branches and accurately assign rewards to local decisions, particularly when the resource budgets increase. This highlights the critical role of our Structured REINFORCE algorithm in enabling precise credit assignment and effective resource allocation.

These results emphasize that FM Programming is not only cost-efficient but also highly adaptive and scalable. Its ability to generalize across both binary and open-form VQA tasks suggests strong potential for real-world deployment in latency- and resource-constrained scenarios.

## 6 Related Works

### 6.1 Multi-modal Reasoning

Multi-modal reasoning tasks, such as visual question answering, require integrating vision and language to address complex queries. End-to-end models, powered by large pre-trained transformers (Deitke et al., 2024; Bai et al., 2025), achieve high accuracy but often come with significant computational costs and limited interpretability. In contrast, modular approaches like ViperGPT (Surís et al., 2023) and VisProg (Gupta & Kembhavi, 2023) decompose complex tasks into smaller, executable programs, enhancing flexibility and interpretability by dynamically composing models based on task needs. Compared to the existing methods that rely on fixed programs, our work extends this modular paradigm with a particular

focus on cost-efficiency, introducing dynamic FM backend selection to handle varying input complexities for achieving the Pareto frontier between resource and performance.

### 6.2 Foundation Model Inference Cost Optimization

Optimizing foundation models, such as LLMs and MLLMs, is essential for deployment in resource-constrained environments. Common techniques include distillation (Hinton et al., 2015; Gu et al., 2023) and quantization (Gholami et al., 2022; Lin et al., 2024) that operate at the model level, and speculative decoding (Leviathan et al., 2023; Miao et al., 2023) and mixture-of-experts routing (Fedus et al., 2022; Huang et al., 2024) that operate at the token level. In contrast, our approach works at the task level, allowing dynamic resource allocation across multiple subtask backends, making it complementary to these strategies.

More recently, Compound AI systems have emerged, leveraging multiple model backends to optimize inference costs. Among these, LLM routing selects the best model for a query to balance cost and quality (Lu et al., 2023; Hu et al., 2024); Cascading uses a series of models, starting with simpler ones for easy tasks and escalating to complex ones as needed (Chen et al., 2023; Nie et al., 2024). Although effective, these methods generally lack integration with task-specific reasoning structures. In contrast, our approach combines task decomposition with dynamic backend selection in FM programs, enabling resource allocation that adapts to both input complexity and task structures for cost saving.

## 7 Conclusion

We introduced the first framework to use foundation model programs in optimizing the trade-off between task performance and resource consumption. By decomposing complex, multi-modal reasoning tasks into programs with control flow and dynamically selecting FM backends in online setups, our method can navigate the Pareto frontier of trade-offs between resource use and performance in real-time, in a way that exploits both task and input structure. As demonstrated in our experiments, this strategy can deliver substantial computational savings with minimal performance degradation.

Many directions of future work remain open. The idea of using FMPs for resource-efficient inference goes beyond the tasks we considered; in particular, we believe they can be deployed in broader agentic and multi-agent applications. Another direction is to explore more advanced program synthesis techniques, such as ones that jointly learn program structure and configurations.

## Limitations

Our framework's performance relies on several assumptions: (a) The initial program structure must be correct – an illogical workflow will lead to sub-optimal performance, regardless of backend allocation; (b) The policy is constrained by the capabilities of the available FM backends. If the backend pool is too weak for a given subtask, even optimal routing may not achieve the desired accuracy, as shown in Appendix Figure 9; (c) The online learning algorithm requires immediate ground-truth feedback, which may not be available in real-world settings; (d) We also do not explicitly model error propagation, where an early mistake from a lightweight module can impact downstream steps.

## Acknowledgement

We thank the anonymous reviewers for their valuable comments and suggestions in enhancing the paper. This research was supported by the NSF under grant numbers CCF-1918651, 2008240, 2131294, and 2212557, NSF PPoSS award #2316161, ARO award #W911NF-21-1-0009, DARPA award #HR00112320018, NIH CTSA award #UM1TR004906, and US DOT Tier-1 UTC CYBER-CARE grant #69A3552348332.

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

## A    No-Regret Guarantee for Structured REINFORCE

We establish that our proposed structured REINFORCE algorithm achieves a no-regret guarantee in an online learning setting. Before presenting the main theorem, we outline the key assumptions that underpin our analysis:

**Assumption 1** (Bounded Rewards). *For all time steps t, configurations $\vec{v}_t \in V$, and inputs $x_t \in \mathcal{D}$, the reward satisfies $R(\vec{v}_t, x_t, y_t) \in [R_{\min}, R_{\max}]$, where $R_{\max} - R_{\min} < \infty$.*

**Assumption 2** (Policy Expressiveness). *For each input $x_t$, there exists an optimal configuration $\vec{v}_{x_t}^* \in V$ that maximizes $R(\vec{v}, x_t, y_t)$. Moreover, the policy class is sufficiently expressive such that there exist parameters $\{\theta_{k_i}^*\}_{i=1}^N$ for which $\pi_{k_i}(m_{k_i,j_i}^* \mid x_t; \theta_{k_i}^*) \approx 1$, where $m_{k_i,j_i}^*$ is the optimal backend for $f_{k_i}$ in $\vec{v}_{x_t}^*$.*

**Assumption 3** (Sufficient Exploration). *The algorithm employs Thompson Sampling with an exploration parameter $\nu > 0$, ensuring that every backend $m_{k_i,j_i} \in M_{k_i}$ has a non-zero probability of being sampled at each time step t.*

**Assumption 4** (Convergence of Policy Gradient). *The learning rate $\eta_t$ is set to $1/\sqrt{t}$, and the policy parameterization (e.g., softmax over $M_{k_i}$) guarantees that gradient updates converge to a near-optimal policy (Agarwal et al., 2021).*

**Assumption 5** (Stationary Input Distribution). *Inputs $x_t$ are drawn independently and identically distributed (i.i.d.) from a fixed distribution $\mathcal{D}$, ensuring a consistent optimal policy over time.*

With these assumptions in place, we can formally state the main result:

**Theorem 1.** *Under Assumptions 1–5, the structured REINFORCE algorithm is no-regret, meaning that the average regret satisfies:*

$$\frac{\gamma_T}{T} \to 0 \quad as \quad T \to \infty,$$

*both in expectation and with high probability, where the regret $\gamma_T$ is defined as:*

$$\gamma_T = \max_{\vec{v} \in V} \sum_{t=1}^{T} R(\vec{v}, x_t, y_t) - \sum_{t=1}^{T} R(\vec{v}_t, x_t, y_t).$$

We now prove this theorem, showing that the algorithm's regret diminishes over time. The proof proceeds by defining the regret, analyzing the algorithm's behavior under the assumptions, and bounding the regret both in expectation and with high probability.

**Proof of Theorem 1**

We start with giving the formal definition of regret $\gamma_T$, which measures the cumulative difference between the maximum achievable reward and the algorithm's actual reward over $T$ steps:

$$\gamma_T = \max_{\vec{v} \in V} \sum_{t=1}^{T} R(\vec{v}, x_t, y_t) - \sum_{t=1}^{T} R(\vec{v}_t, x_t, y_t).$$

Our goal is to show that the average regret, $\frac{\gamma_T}{T}$, approaches zero as $T \to \infty$. To simplify the analysis, we use a stronger benchmark: the optimal context-dependent configuration $\vec{v}_{x_t}^*$ that maximizes $R(\vec{v}, x_t, y_t)$ for each $x_t$. Since $\max_{\vec{v} \in V} R(\vec{v}, x_t, y_t) \leq R(\vec{v}_{x_t}^*, x_t, y_t)$, we have:

$$\gamma_T \leq \sum_{t=1}^{T} R(\vec{v}_{x_t}^*, x_t, y_t) - \sum_{t=1}^{T} R(\vec{v}_t, x_t, y_t).$$

This upper bound focuses the proof on the gap between the optimal and achieved rewards per step.

The structured REINFORCE algorithm updates sub-policy parameters $\theta_{k_i}$ using policy gradients. The expected cumulative reward is:

$$\mathcal{J}(\pi) = \mathbb{E}_{\pi} \left[ \sum_{t=1}^{T} R(\vec{v}_t, x_t, y_t) \right],$$

with the gradient for each sub-policy:

$$\nabla_{\theta_{k_i}} \mathcal{J}(\pi_{k_i}) = \sum_{t=1}^{T} \mathbb{E}_{j_i^* \sim \pi_{k_i}(\cdot | x_t)} \left[ \nabla_{\theta_{k_i}} \log \pi_{k_i}(m_{k_i, j_i^*} | x_t; \theta_{k_i}) \cdot r_{k_i, j_i^*} \right].$$

The algorithm approximates this gradient with a single sample, updating parameters as:

$$\theta_{k_i, t+1} = \theta_{k_i, t} + \eta_t \nabla_{\theta_{k_i}} \log \pi_{k_i}(m_{k_i, j_i^*} | x_t; \theta_{k_i, t}) r_{k_i, j_i^*},$$

where $\eta_t = 1/\sqrt{t}$ (Assumption 4), and the sub-reward is defined as:

$$r_{k_i, j_i^*} = -\lambda \mathcal{C}(m_{k_i, j_i^*}) - \frac{1}{N} \mathcal{L}(p(x_t | \vec{v}_t), y_t).$$

Assumptions 2 (expressive policy class) and 3 (sufficient exploration via Thompson Sampling) ensure that each sub-policy $\pi_{k_i}$ converges to the optimal sub-policy $\pi_{k_i}^*$, where $\pi_{k_i}^*(m_{k_i, j_i}^* | x_t) \approx 1$ for the optimal backend $m_{k_i, j_i}^*$ in $\vec{v}_{x_t}^*$. Thus, the overall policy $\pi_t$ converges to the optimal policy $\pi^*$, leading to $\mathbb{E}_{\pi_t}[R(\vec{v}_t, x_t, y_t)] \to R(\vec{v}_{x_t}^*, x_t, y_t)$ as $t \to \infty$.

Define the per-step regret as:

$$\rho_t = R(\vec{v}_{x_t}^*, x_t, y_t) - R(\vec{v}_t, x_t, y_t),$$

with expected value:

$$\mathbb{E}[\rho_t] = \mathbb{E}_{x_t \sim \mathcal{D}} \left[ R(\vec{v}^*_{x_t}, x_t, y_t) - \mathbb{E}_{\vec{v}_t \sim \pi_t} [R(\vec{v}_t, x_t, y_t)] \right].$$

Since $\pi_t \to \pi^*$, we have $\mathbb{E}[\rho_t] \to 0$. The total expected regret is:

$$\mathbb{E}[\gamma_T] \le \sum_{t=1}^{T} \mathbb{E}[\rho_t].$$

Given $\eta_t = 1/\sqrt{t}$ and standard policy gradient convergence (Assumption 4), we bound:

$$\mathbb{E}[\gamma_T] \le C\sqrt{T},$$

for some constant $C$ based on the reward bounds and policy parameters (Zinkevich, 2003; Ban et al., 2021). Thus, the average expected regret satisfies:

$$\frac{\mathbb{E}[\gamma_T]}{T} \le \frac{C}{\sqrt{T}} \to 0 \quad \text{as} \quad T \to \infty,$$

establishing no-regret in expectation.

Now, we extend this to show that the actual regret $\gamma_T$ converges similarly with high probability. By Assumption 1, rewards are bounded, so $\rho_t \in [-(R_{\max} - R_{\min}), R_{\max} - R_{\min}]$, and let $B = R_{\max} - R_{\min}$. Since $x_t$ are i.i.d. (Assumption 5) and $\vec{v}_t$ are sampled independently given $x_t$ and $\pi_t$, the $\rho_t$ are independent. Applying Hoeffding's inequality to $\gamma_T = \sum_{t=1}^{T} \rho_t$:

$$\mathbb{P}\left( |\gamma_T - \mathbb{E}[\gamma_T]| \ge \epsilon \right) \le 2 \exp\left( -\frac{2\epsilon^2}{TB^2} \right).$$

Set $\epsilon = \delta T$, so:

$$\mathbb{P}\left( |\gamma_T - \mathbb{E}[\gamma_T]| \ge \delta T \right) \le 2 \exp\left( -\frac{2\delta^2 T}{B^2} \right).$$

This probability approaches 0 exponentially as $T \to \infty$. Thus, with probability at least $1 - 2 \exp\left( -\frac{2\delta^2 T}{B^2} \right)$, which approaches 1 as $T$ grows, we have:

$$\gamma_T < \mathbb{E}[\gamma_T] + \delta T \le C\sqrt{T} + \delta T.$$

Dividing by $T$, we find:

$$\frac{\gamma_T}{T} < \frac{C}{\sqrt{T}} + \delta.$$

For any $\epsilon > 0$, choose $\delta = \frac{\epsilon}{2}$ and $T > \left( \frac{2C}{\epsilon} \right)^2$, so $\frac{C}{\sqrt{T}} < \frac{\epsilon}{2}$, yielding:

$$\frac{\gamma_T}{T} < \epsilon,$$

with probability approaching 1. Hence, $\frac{\gamma_T}{T} \to 0$ with high probability, completing the no-regret proof.

## B  Detailed Experimental Setups

All experiments are conducted on a single machine equipped with 8 NVIDIA A40 GPUs (48GB memory each), running CUDA 12.4. Detailed package versions are listed in the environment file available at `GitHub`. Hyperparameter settings are specified in the experimental configuration files for `binary VQA` and `open-form VQA`.

For the offline code generation, we use a prompt format similar to ViperGPT, including the DSL, the user query, and a one-shot example. In practice, the generated programs are >90% correct, and errors are easily caught during dry runs or audits. A full prompt example is also available at `GitHub`.

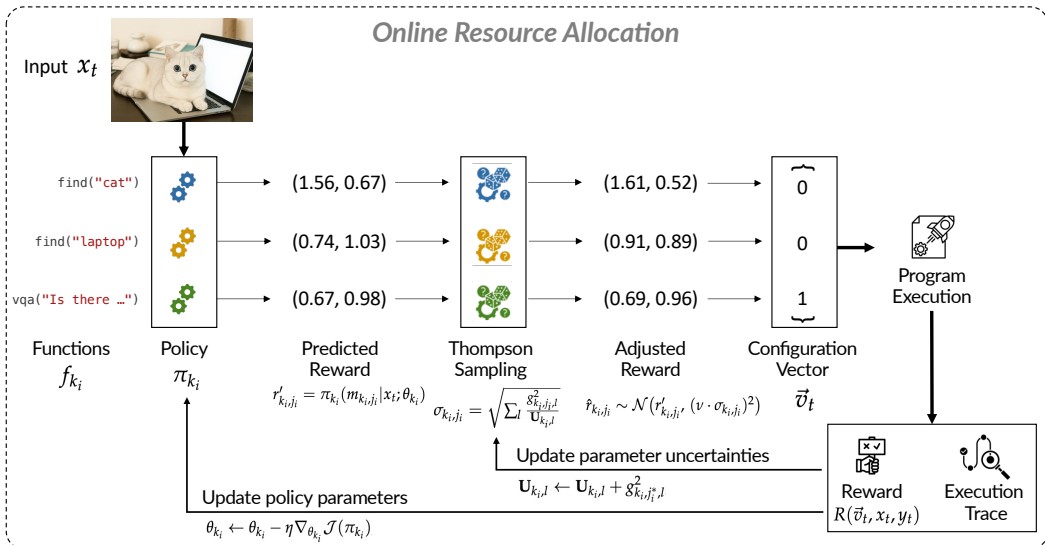

Figure 5: A walk-through example showing how an FMP is compiled and routed with online resource allocation in Algorithm 1. Here we look at a specific input $x_t$ for illustration but the user inputs actually arrive continuously in a stream.

The current FM backend is set up with VLLM (Kwon et al., 2023), Huggingface Hub, MMDetection (Chen et al., 2019), and AgentLego (AgentLego, 2023). Our framework supports a flexible backend construction with any open-source or closed-source API-based models. However, due to the unavailability of computational costs of closed-source models, such as GPT and Gemini series models, we do not include them in our FM backend during experiments.

For open-form VQA experiments, we consistently use the same system prompt for MLLMs:

---

**System Prompt 1**

```
Keep your answer short. Try to answer within 3 words. For numerical answers,
use number digits (e.g., 5 instead of five), and returns the number only. If
there are multiple answers, separate them with a comma (e.g., cat, dog). If you
find the question unanswerable based on the image content, output "N/A". For
example, if the image content is irrelevant to the question, or the content in
the image does not fully and clearly match all the entities, humans, attributes,
spatial, logical, and numerical constraints in the question, output "N/A".
```

---

## C  Detailed Benchmark Construction and Evaluation

### C.1  Streaming Binary VQA

**Benchmark Construction.**  Our dataset is constructed from COCO (Lin et al., 2014), selecting captions that require multi-object compositional reasoning with spatial, logical, or numerical constraints. For each query, we prepare a set of more than 2,000 images, sampled based on the similarity of their captions to the query. The system must output a binary decision (*yes/no*) for each image indicating whether it satisfies the compositional query. To enforce structured reasoning, we leverage an LLM[2] to generate FM programs in a predefined DSL. These programs decompose the query into discrete reasoning steps, guiding the selection of foundation models for subtask execution. A detailed pipeline for benchmark construction is illustrated in Figure 6.

---

[2]We use GPT-4o (Hurst et al., 2024) as the LLM throughout this section unless otherwise specified.

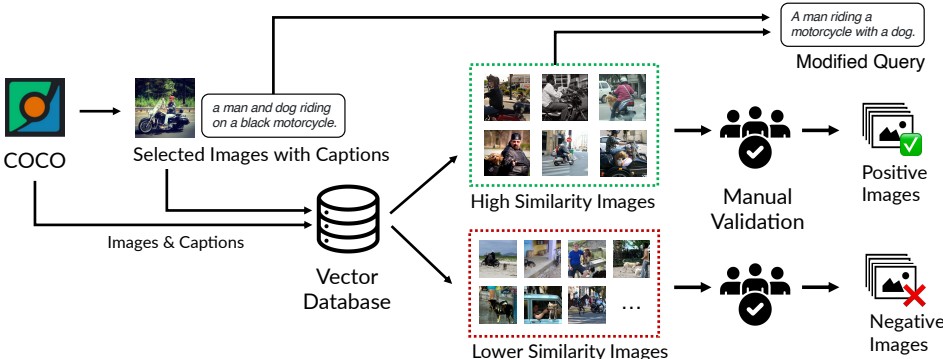

Figure 6: Benchmark construction pipeline for the Streaming Binary VQA dataset.

**Annotation and Verification.** Eleven human annotators validate the correctness of the neurosymbolic programs and the image labels, ensuring that each program aligns with the intended reasoning process and each image is correctly labeled against the compositional constraints. The final dataset consists of 33 queries covering three primary reasoning types (note that a query may fall into multiple types):

- **Spatial Reasoning** (20 queries): These queries require understanding and interpreting the spatial relationships between objects or people in an image. They often describe where things are located relative to each other, *e.g.*, "*Is there a person riding a bicycle next to a bus on the street*".
- **Logical Reasoning** (15 queries): These queries involve conditions, attributes, or combinations that require deductive thinking. The model needs to process logical relationships, such as inclusion, exclusion, or conjunction, *e.g.*, "*Are there people riding bikes, scooters, or motorcycles while holding or using umbrellas?*".
- **Numerical Reasoning** (9 queries): These queries test the ability to understand and count quantities or numbers in a scene. They often specify exact counts or comparisons, *e.g.*, "*Are there at least four horses on a beach*".

**Evaluation.** Since the benchmark is highly imbalanced, with a positive-to-negative ratio of around 1:100, task performance is measured using accuracy, recall, precision, and F1-score.

### C.2 Streaming Open-form VQA

**Benchmark Construction.** To enable evaluation on more complex open-form questions, we construct a dataset comprising 50 queries and 25,000 images, spanning five distinct reasoning categories. To ensure the validity and diversity of these queries, we randomly sample them from established benchmark datasets. Specifically, spatial queries are drawn from the GQA dataset (Hudson & Manning, 2019), focusing on queries labeled as relS and categoryRelS. Logical queries are also sampled from GQA, targeting the detailed types twoCommon, twoSameMaterialC, twoDifferentC, and twoDifferent. Numerical queries are selected from the A-OKVQA dataset (Schwenk et al., 2022), while comparative and external knowledge queries are sourced from OKVQAS3 (Jain et al., 2021). To enhance clarity and naturalness, some of the queries are manually rewritten.

Each query is associated with 500 generated images. The image generation pipeline begins with an LLM generating a set of 10 possible answers for each query, proposing potential scene setups along with 3 additional objects, and constructing detailed image descriptions. These descriptions are then used to prompt a diffusion model[3] for image generation.

To assess model robustness and reasoning precision, we incorporate unanswerable images that are visually coherent but semantically invalid with respect to the query. These include

---

[3]We use FLUX.1-dev (Black Forest Labs, 2023) as the diffusion model for image generation.

| | | VQA v2.0 | GQA | CLEVR | A-OKVQA | Streaming VQA Binary | Streaming VQA Open-form |
|---|---|---|---|---|---|---|---|
| **Query** | Multiple Objects | ✗ | ✓ | ✓ | ✓ | ✓ | ✓ |
| | Spatial Reasoning | ✓ | ✓ | ✓ | ✓ | ✓ | ✓ |
| | Logical Reasoning | ✓ | ✓ | ✓ | ✓ | ✓ | ✓ |
| | Numerical Reasoning | ✓ | ✓ | ✓ | ✓ | ✓ | ✓ |
| | Comparative Reasoning | ✓ | ✓ | ✓ | ✓ | ✗ | ✓ |
| | External Knowledge | ✗ | ✗ | ✗ | ✓ | ✗ | ✓ |
| **Image** | Source | COCO | COCO & Flickr | Synthetic | COCO | COCO | Generation |
| | Unanswerable Images | ✗ | ✗ | ✗ | ✗ | ✗ | ✓ |
| **Scale** | # Queries | 1.1M | 22M | 999,968 | 24,903 | 33 | 50 |
| | # Images | 200K | 113K | 100,000 | 23,692 | 66,279 | 25,000 |
| | **# Image(s) per query** | 2 | 1 | 1 | 1 | **>2000** | **500** |

Table 1: Comparing to the existing VQA benchmarks (Goyal et al., 2017; Hudson & Manning, 2019; Johnson et al., 2017; Schwenk et al., 2022), Streaming VQA is the first that provides a sequence of images for each query.

both unrelated (random) images and images that are intentionally crafted to closely resemble answerable cases, making them more difficult to distinguish, as illustrated in Figure 2. This setup challenges models not only to infer the correct answer when possible but also to recognize when a question cannot be answered from the image.

Additionally, for each query, we also synthesize the corresponding FM program in the predefined DSL, providing a structured decomposition of the reasoning process.

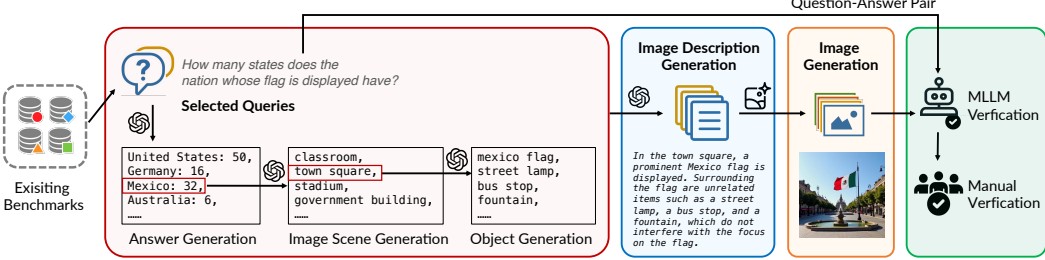

Figure 7: Benchmark construction pipeline for the Streaming Open-form VQA dataset.

**Annotation and Validation.** A two-step validation process is employed. First, a multimodal LLM, GPT-4o-mini (OpenAI, 2024), verifies each image by generating an answer and comparing it to the expected answer. Only images where the MLLM's response matches the assigned answer are retained. Then, human evaluators verify a random image subset, achieving approximately 93% accuracy. The final benchmark encompasses five reasoning types:

- **Spatial Reasoning** (13 queries): These questions require understanding the spatial relationships between objects within a scene, *e.g.*, "*What is in the jar to the left of the juice?*".
- **Logical Reasoning** (9 queries): This category involves applying conditions, rules, or filters to identify specific objects or answer complex queries, *e.g.*, "*What is the black object on the desk that is not electronic?*", "*How many people are wearing both glasses and a hat?*".
- **Numerical Reasoning** (11 queries): This category requires counting, comparing numbers, or calculating quantities based on visual information, *e.g.*, "*How many extra bottles of beer do we need to make it a half dozen?*".
- **Comparative Reasoning** (11 queries): These questions involve evaluating two or more objects in terms of their attributes, such as size, height, quantity, or quality, *e.g.*, "*Which bottle is taller, the left one or the right one?*".
- **External Knowledge Reasoning** (6 queries): These questions rely on information that extends beyond what is immediately visible in the image, often drawing on common sense or factual knowledge, *e.g.*, "*The fruit in the picture is a good source of what vitamin?*", "*How many states are there in the country whose flag is being displayed?*".

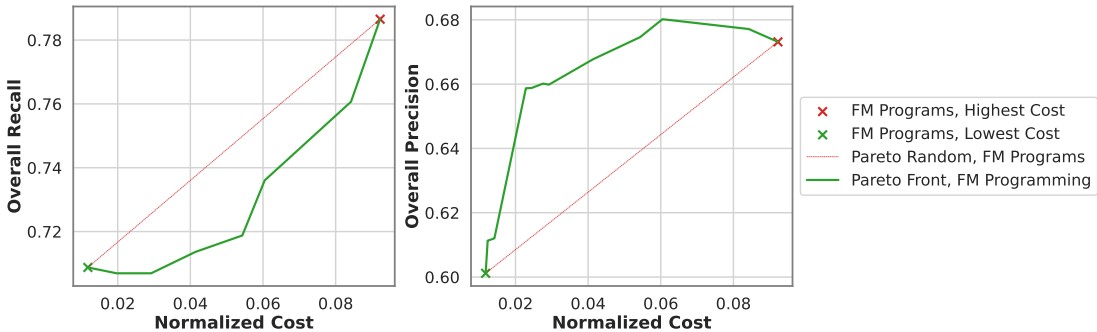

Figure 8: Pareto front of precision and recall scores on the Streaming Binary VQA benchmark. The policy prioritizes precision over recall as the cost budget increases.

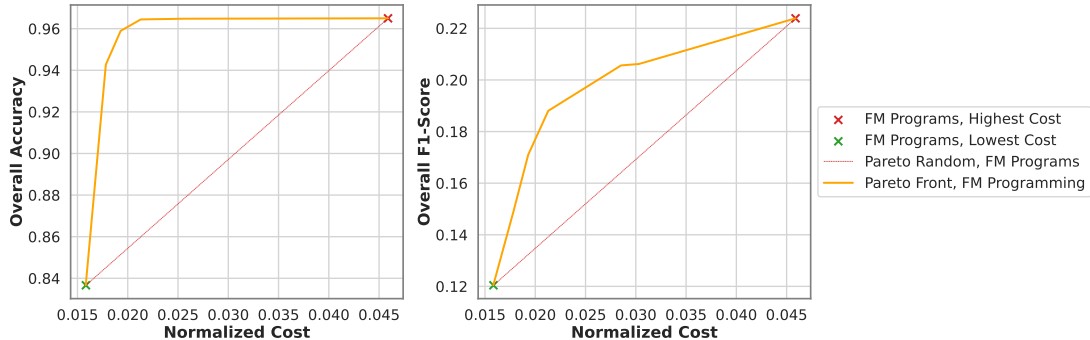

Figure 9: Additional results on the Streaming Binary VQA benchmark with different FM backends. Costs are normalized based on the inference costs of Qwen2.5-VL 72B. The backends include GLIP tiny (231M) and base (430M) models (Li et al., 2022b) for object detection, OFA base (182M) (Wang et al., 2022) and BLIP-2 OPT-2.7B (3.745B) (Li et al., 2023) for language-vision understanding. FM programming outperforms the baseline with better cost-performance trade-offs, demonstrating its adaptability across backend setups.

**Evaluation.** Performance for the streaming VQA task is evaluated using exact match accuracy, measuring the proportion of questions answered correctly without partial credit.

