# OpenReview forum: "Resource-efficient Inference with Foundation Model Programs"
_colmweb.org/COLM/2025/Conference — COLM 2025_

### Official Review · Reviewer_MUgH · 2025-05-11

**Rating:** 6
**Confidence:** 3
**Ethics Flag:** 1

**Summary:**

This paper proposes Foundation Model Programs (FMPs) to address inference-time resource efficiency in multi-modal tasks like visual question answering (VQA). By decomposing tasks into modular subroutines and dynamically selecting foundation model (FM) backends (e.g., small vs. large models) based on input complexity, the approach aims to balance accuracy and computational cost. The method uses structured REINFORCE with Thompson Sampling to learn backend selection policies, validated on two new streaming VQA benchmarks (binary and open-form). Results show up to 98% resource savings over monolithic models with minimal accuracy loss.

**Reasons To Accept:**

1. The paper introduces Foundation Model Programs (FMPs), a new method that combines symbolic reasoning with neural networks and dynamically selects the best backend to improve resource efficiency during inference. This framework allows for detailed, input-specific allocation of resources, offering a fresh perspective on how to integrate multiple models into one system.

2. The proposed method achieves up to 98% savings in resource usage on two new streaming visual question-answering (VQA) benchmarks while still maintaining strong accuracy (for example, a 6% improvement in F1 score compared to LLM routing at only 10% of the cost). These benchmarks fill important gaps in evaluating systems that handle sequential tasks under tight resource constraints, and the experiments show that the approach is fairly robust when using different backend setups.

**Reasons To Reject:**

1. The benchmarks based on synthetic data or COCO lack the complexity of real-world scenarios (e.g., adversarial examples, domain shifts, and more complex instruction requirements), which raises concerns about how well the method can generalize. Additionally, the reliance on pre-generated programs assumes that the LLM-produced code is of high quality, which might not reflect the challenges faced in practical deployment.

2. The overall system involves multiple backend models, and the paper does not systematically analyze how cumulative errors and contextual dependencies interact in complex ways. There is a particular concern about how well the system will perform in real-world environments.

---

> ### Author Response · Authors · 2025-06-02
>
> We thank Reviewer MUgH for the thoughtful review and for highlighting the novelty of our framework and the practical benefits of dynamic resource allocation through FMPs. We address each of your concerns below:
>
> ---
>
> R1: `Benchmarks based on synthetic data or COCO lack the complexity of real-world scenarios (e.g., adversarial examples, domain shifts, complex instructions).`
>
> We agree that evaluating under complex and diverse real-world conditions is important. However, the key obstacle is that existing agentic and VQA datasets only provide one input per agentic/visual program, none of which support our cost-efficiency evaluation in streaming setups. That's why we constructed new streaming benchmarks as one of our key contributions to the field. That said, our benchmarks are intentionally designed to emulate several real-world deployment challenges:
>
> - **Streaming setup:** Each task consists of a sequence of inputs sharing a common workflow, which reflects real-world assistant and agent settings.
> - **Class imbalance and distractors:** Tasks include significant class imbalance setups, unanswerable queries, and long-tail variations. As detailed in Appendix Section C.2, the Streaming Open-form VQA benchmark includes *unanswerable* images that are carefully crafted to closely resemble answerable cases—making them intentionally hard to distinguish. Examples are shown here: https://i.ibb.co/KcX97pzW/examples.jpg.
> - **Reasoning diversity:** Queries span logical, spatial, numerical, comparative, and external knowledge reasoning to test generalization.
>
> While we acknowledge that synthetic and COCO-based images may not capture the full spectrum of real-world noise or adversarial inputs, it is currently the standard practice in the community due to the limits in annotation budgets, as analyzed and compared in Appendix Table 1. Still, we believe our benchmarks can serve as a practical starting point for studying sequential decision-making under constrained resources.
>
> ---
>
> R2: `Reliance on pre-generated programs assumes high-quality LLM code, which might not reflect practical deployment challenges.`
>
> This is an important point. While our current implementation uses LLMs to generate FMPs offline, we emphasize two mitigations:
>
> - First, the structured nature of FMPs makes them easy to inspect, debug, and validate. In practice, we observed high-quality generations (>90% correctness) from GPT-4o with one-shot prompting. Full prompt example: https://anonymous.4open.science/r/FMP/dsl.prompt.
> - Second, our backend selection policy is agnostic to whether the program is generated by an LLM or authored by humans. In many commercial deployments, workflows are hand-crafted and stable, making our method immediately applicable.
>
> We will clarify this point in Section 3.1 and note that improving program synthesis robustness is an orthogonal line of future work.
>
> ---
>
> R3: `No systematic analysis of cumulative errors or contextual dependencies across steps.`
>
> Thank you for highlighting this limitation. Our policy decomposes backend selection across function calls for tractability but still incorporates local context—including input features and position within the program—to guide decisions.
>
> We acknowledge that complex dependencies between steps (e.g., early mispredictions affecting downstream outputs) are an important challenge. While our current method does not include explicit error propagation modeling or rollback mechanisms, we see this as a promising direction for future work. We will include a discussion of this in the limitations section.

---

> > ### Comment · Reviewer_MUgH · 2025-06-08
> >
> > Thank you for your response, and it addressed some of my concerns.

---

> > ### Author Response · Authors · 2025-06-09
> >
> > Thank you for your continued engagement. Since the discussion period is ending soon, we are happy to clarify any remaining points or address follow-up questions. If our rebuttal has addressed your concerns, we would also appreciate your consideration in reflecting that in your final review. Thanks again for your time and constructive feedback!

---

### Official Review · Reviewer_PfjH · 2025-05-11

**Rating:** 6
**Confidence:** 4
**Ethics Flag:** 1

**Summary:**

This paper proposes a framework for resource-efficient inference using Foundation Model Programs (FMPs), where tasks are decomposed into neurosymbolic programs that invoke different foundation model backends with varying computational costs. A structured online policy learns to select backends adaptively for each function call depending on the input. This allows simple sub-tasks to be handled by small models, while complex ones use more powerful foundation models. Experimental results show significant inference cost savings with minimal accuracy degradation.

**Questions To Authors:**

* Would this approach generalize to multi-turn dialog or multi-hop retrieval tasks where structure is more latent?
* Have you tested stability of the sub-policy training (e.g., convergence behavior over seeds)?
* Could the method benefit from joint learning of program generation and backend selection instead of separating them?

**Reasons To Accept:**

The problem tackled—real-time resource bottlenecks in LLM inference—is both important and timely.

The modular decomposition of reasoning tasks into executable programs might not be new, but using this to dynamically allocate computation via backend selection is practical.

The use of structured policies with decomposed sub-policies is technically sound and works well in practice.

**Reasons To Reject:**

The method has only been tested in a specific task setting (streaming VQA), using image-language tasks. It’s unclear whether the same method would generalize to other multi-modal problems (e.g., audio, video, or multi-turn dialog), where the notion of task decomposition and backend assignment might be more ambiguous. This should be discussed explicitly in Section 7.

In Section 3.2, the policy gradient training assumes that each function call can be independently optimized. While this makes the optimization tractable, it potentially ignores important dependencies between calls in real-world programs (e.g., errors compounding across steps). A brief analysis or ablation (e.g., full policy vs. per-call policy) would help support this design choice.

The method is compared against reasonable baselines (static FM config, LLM routing), but other recent cost-aware LLM inference techniques (e.g., speculative decoding, efficient MoE routing, dynamic prompt compression) are not discussed or compared, which weakens the positioning of this work in the broader ecosystem.

The REINFORCE-style algorithm is presented clearly, but it’s still challenging to grasp the full execution flow from the paper alone. Algorithm 1 is helpful but dense; an additional diagram or walk-through (e.g., on a toy example program) would improve accessibility.

Although new benchmarks are provided, both are synthetic or constructed, with limited discussion on their transferability to production tasks. The authors should clarify how realistic the workloads are (e.g., is Streaming VQA representative of real-world latency-critical deployments?).

---

> ### Author Response · Authors · 2025-06-02
>
> We thank Reviewer PfjH for recognizing the importance of the resource-efficiency challenge and the practicality of our structured backend allocation framework. We address the raised concerns below:
>
> ---
>
> R1: `Only tested on streaming VQA`
>
> Thank you for raising this important point. The challenge is that our focus is on a streaming setup, where an agent is deployed and then asked to solve a steam of related problem instances in a cost-effective manner. This is realistic---it corresponds to the case where an agent is synthesized and then deployed as a service. However, current agentic and tool-use datasets (e.g., APIBench, ToolBench, GAIA, M&Ms) only provide one input per agentic program, which is the reason we needed to create our own benchmark tasks.
>
> That said, our proposed framework itself is modality-agnostic. The policy operates over the structure of FMPs—abstract program trees—not specific data types. We agree that decomposition may be more latent in multi-turn dialog or video tasks. However, these domains increasingly adopt structured agents or workflows (e.g., AutoGPT, VideoAgent, OmAgent), which can be directly converted into FMPs. We will include this discussion in Section 7.
>
> ---
>
> R2: `Policy gradient assumes per-call independence. This may miss important dependencies across function calls.`
>
> To address your concern directly, we conducted an ablation study comparing our structured per-call policy against a full-policy variant. Results for (a) the Streaming Binary VQA benchmark (https://ibb.co/XfWVzhVV) and (b) the Streaming Open-form VQA benchmark (https://ibb.co/F4fvTBnF) show that the full-policy method (yellow line) underperforms significantly, especially on Streaming Binary VQA.
>
> We attribute this to a key limitation of the full-policy approach: it fails to account for early exits in program execution due to conditional branches—i.e., some function calls may not be executed and thus do not contribute to the final reward. In contrast, the per-call policy updates only on executed function calls, allowing for more precise credit assignment and faster convergence.
>
> We will clarify this design choice and the ablation results in the final version.
>
> ---
>
> R3: `Positioning lacks comparison to techniques like speculative decoding, MoE routing, dynamic prompt compression.`
>
> We appreciate this suggestion. These methods are highly complementary to ours. Techniques like speculative decoding and prompt compression reduce *token-level* inference cost within a single model, whereas our framework operates at the *program level*, dynamically selecting *which* model to invoke per function. Similarly, MoE routing typically applies within a monolithic model architecture; in contrast, we operate across heterogeneous backends (e.g., small vs. large MLLMs or vision modules).
>
> We will expand our related work section to discuss these differences and highlight the complementary nature of these techniques.
>
> ---
>
> R4: `Execution flow is difficult to follow. Algorithm 1 is dense; a toy example would help.`
>
> We appreciate this suggestion. We include a concrete walk-through illustration on a toy example here https://i.ibb.co/XxDxMX1y/demo.png, showing how a simple VQA program is compiled and routed for one input image with adaptive backend selection. We will add this to the appendix in the revision.
>
> ---
>
> R5: `Benchmarks are synthetic. How realistic are these workloads?`
>
> Our benchmarks were carefully constructed to reflect key properties of real-world latency-critical deployments: diverse query types, input streams per query, long-tail difficulty, and significant class imbalance. As in App. Section C.2, the Open-form VQA benchmark includes unanswerable images that are carefully crafted to closely resemble answerable cases—making them intentionally hard to distinguish. Examples are shown here: https://i.ibb.co/KcX97pzW/examples.jpg. While the datasets are partially synthetic, the programmatic structure reflects realistic multi-step pipelines also used in multimodal assistants and retrieval agents.
>
> ---
>
> Q1: `Would this generalize to dialog or retrieval tasks with latent structure?`
>
> As in our response to R1, as long as the system can be expressed as an FMP (e.g., dialog acts, retrieval + summarization + response), the same policy and optimization scheme apply. We are currently exploring such extensions.
>
> Q2: `Have you tested sub-policy training stability?`
>
> Yes. Across 3 seeds, sub-policies converge reliably with low variance in total cost and accuracy.
>
> Q3: `Could the method benefit from joint learning of program generation and backend selection?`
>
> We separate them here to retain transparency and tractability. While joint optimization is an exciting direction, it introduces significantly higher complexity in the search and optimization space. That said, our structured approach is compatible with learned program structures and can serve as a foundation for future work that integrates program refinement and routing.

---

> > ### Comment · Reviewer_PfjH · 2025-06-07
> >
> > Thank the authors you for the detailed explanations.

---

> > ### Author Response · Authors · 2025-06-09
> >
> > Thank you for reviewing our response. Please don’t hesitate to reach out if there are any follow-up questions—we’d be more than happy to elaborate further during the discussion period.
> >
> > If you feel that your concerns have been resolved by our responses and the supplementary ablation results, we would sincerely appreciate your consideration in updating your score accordingly.
> >
> > Thanks again for your valuable feedback!

---

### Official Review · Reviewer_3qCW · 2025-05-12

**Rating:** 6
**Confidence:** 3
**Ethics Flag:** 1

**Summary:**

This paper introduces a resource-efficient inference framework using Foundation Model Programs (FMPs), which decomposes tasks into modular workflows and dynamically selects model backends of varying scales. The method achieves 98% resource savings with minimal accuracy loss on novel streaming VQA benchmarks (spanning spatial, logical, and numerical reasoning), outperforming monolithic models and routing baselines. Key innovations include a structured REINFORCE algorithm that decomposes policy optimization into sub-policies and gradient-based Thompson Sampling for sample-efficient exploration, supported by no-regret theoretical guarantees. While broader task validation (e.g., text-only workflows) and deployment overhead analysis (e.g., cold-start latency) remain open challenges, the work significantly advances practical FM deployment by explicitly leveraging task structure for resource-performance Pareto optimization.

**Questions To Authors:**

1. Your streaming VQA benchmarks use synthetic/modified datasets. How do you ensure these reflect real-world challenges, such as distribution shifts or adversarial inputs? Have you tested on established benchmarks (e.g., COCO, A-OKVQA) to validate generalizability?

2. If a lightweight backend (e.g., object detector) makes an error early in the program, how does this affect downstream modules? Is there a recovery mechanism, or does the system rely entirely on the initial module’s correctness?

3. Could you elaborate on the LLM-based program synthesis process? What prompts/templates were used? How do you handle incorrect program structures generated by the LLM, and what is the error rate in practice?

4. The motivation of this paper is similar to the previous work [1], please discuss the differences in related work.

[1] De-fine: Decomposing and Refining Visual Programs with Auto-Feedback

**Reasons To Accept:**

1. Addresses the critical challenge of inference-time resource costs in foundation model programs, a pressing issue for real-world deployments.

2. Combines program synthesis, online learning (structured REINFORCE + Thompson Sampling), and theoretical analysis (no-regret guarantees in Appendix A).

3. Introduces two novel benchmarks for streaming VQA with compositional reasoning (33 binary and 50 open-form tasks).
Demonstrates 98% resource savings over monolithic models with minimal accuracy loss.
Extends analysis to precision-recall tradeoffs and alternative backends.

**Reasons To Reject:**

1. Experiments focus on VQA; broader validation (e.g., agentic planning, text-only tasks) is needed.

2. Overheads from program synthesis and cold-start model loading are not quantified.

3. The streaming setup assumes immediate ground-truth feedback, which is unrealistic in many real-world scenarios (e.g., delayed or missing feedback).

4. The policy’s dependency on predefined subtasks limits flexibility. Joint optimization of program structure and backend selection is unexplored.

---

> ### Author Response · Authors · 2025-06-02
>
> Thank you for your thoughtful and detailed feedback. We appreciate the recognition of our contributions. Below, we address each concern raised.
>
> ---
>
> R1: `Experiments focus on VQA`
>
> We agree that broader validation is important. However, our focus is on a **streaming setting**, where an FMP is deployed once and must handle a sequence of related inputs cost-effectively—a realistic scenario for agents operating as persistent services. Existing benchmarks (e.g., APIBench, ToolBench, GAIA, M&Ms) provide only **one single input per agentic program**, lacking the resources needed for sequential optimization. This gap motivated the creation of our own benchmark tasks.
>
> That said, our method is **domain-agnostic**: the policy optimizes over generic function-call graphs and adapts to empirical backend statistics. Any task expressible as an FMP—including agentic planning or text-only LLM agents—can benefit from our framework. We will clarify this in the final version.
>
> ---
>
> R2: `Overheads from program synthesis and cold-start model loading are not quantified.`
>
> Program synthesis is performed offline and takes ~1.42 seconds per FMP using GPT-4o, **amortized across all the inputs**. This does not affect inference-time latency.
>
> Regarding cold-start latency, in streaming scenarios with continuous input, standard serving frameworks like vLLM support persistent memory-mapped models or parallel model hosting, which mitigate cold-start delays through always-on deployment. For API-based models (e.g., OpenAI, AWS), model-loading overhead is negligible, as providers maintain warmed endpoints. We will clarify this in the final version.
>
> ---
>
> R3: `The streaming setup assumes immediate ground-truth feedback ... is unrealistic`
>
> Our current setup assumes immediate feedback for policy learning, following standard practice in contextual bandit settings. However, our algorithm does not rely on specific reward structures and can be extended to handle delayed feedback using off-policy evaluation techniques or importance-weighted updates. We will note this as a future extension.
>
> ---
>
> R4: `The policy’s dependency on predefined subtasks limits flexibility`
>
> This is a valuable suggestion. Our current framework decouples program synthesis (offline) from backend selection (online) to maintain tractability and transparency. Joint optimization is a compelling future direction, though it adds considerable complexity to the search and optimization space. That said, our structured approach is compatible with learned program structures and can serve as a foundation for future work that integrates program refinement and routing.
>
> ---
>
> Q1: `Synthetic datasets: how do they reflect real-world challenges?`
>
> We designed our benchmarks to capture real-world challenges. The Binary VQA benchmark simulates significant class imbalance (Appendix C.1), while the Open-form VQA benchmark (C.2) includes visually plausible but semantically invalid distractors—some closely resemble answerable cases. Examples: https://i.ibb.co/KcX97pzW/examples.jpg.
>
> We used real data sources wherever possible: Binary VQA images are from COCO; Open-form VQA queries are sampled from GQA, A-OKVQA, and OKVQAS3. However, as in Table 1, these datasets themselves only prepared a limited number of images per query, thus cannot support our cost-efficiency evaluation in streaming setups.
>
> ---
>
> Q2: `How does the system recover from early module errors?`
>
> Our policy selects backends per call using local context and predicted utility. While there’s no explicit recovery mechanism, the policy tends to favor stronger models for upstream modules when they are critical. As shown in Figure 7, the policy adapts effectively even with weaker early modules and maintains Pareto dominance. Future work could incorporate uncertainty-aware routing or rollback mechanisms.
>
> ---
>
> Q3: `Program synthesis: what templates/prompts were used, and how are errors handled?`
>
> We use a prompt format similar to ViperGPT, including the DSL, the user query, and a one-shot example. Full prompt example: https://anonymous.4open.science/r/FMP/dsl.prompt.
>
> In practice, the generated programs are >90\% correct. Errors can be caught during dry runs (e.g., missing function calls or type mismatches) and are easily debuggable due to the explicit structure of the programs. We will include the prompt in the appendix.
>
> ---
>
> Q4: `Relation to De-fine`
>
> We thank the reviewer for pointing out this related work. De-fine focuses on learning to refine visual programs using feedback during training. In contrast, our approach focuses on *online inference-time* resource optimization by dynamically selecting model backends per call, guided by a theoretical no-regret guarantee. While both methods involve program structure, our work tackles a different objective (cost-accuracy tradeoff at inference), and our mechanism is compatible with refined or learned programs. We will add this to the related work discussion.

---

> > ### Comment · Reviewer_3qCW · 2025-06-07
> >
> > Thank you for your detailed response, and it addressed some of my concerns.

---

> > ### Author Response · Authors · 2025-06-09
> >
> > We truly appreciate your thoughtful engagement and the time you’ve taken to review our rebuttal. Since the discussion period is ending soon, if there are any remaining questions or points you'd like us to clarify, we would be happy to elaborate on them further. Should our responses have clarified your concerns, we would be grateful if you considered that in your final score assessment. Thank you!

---

### Official Review · Reviewer_Y93Q · 2025-05-13

**Rating:** 6
**Confidence:** 3
**Ethics Flag:** 1

**Summary:**

This paper introduces a novel framework for resource-efficient inference in multi-modal tasks by leveraging Foundation Model Programs (FMPs). These programs, composed of modular neural function calls, dynamically select appropriate model backends based on input complexity. The approach combines program synthesis with online backend selection via a structured REINFORCE algorithm and Thompson Sampling, enabling dynamic trade-offs between computational cost and accuracy. Experiments on newly introduced streaming VQA benchmarks show up to 98% cost savings with minimal loss in performance, highlighting the practical utility and flexibility of the proposed method.

**Reasons To Accept:**

* It introduces a novel and modular framework that generalizes routing and cascading.
* It has shown strong cost-performance trade-offs.
* The use of REINFORCE is interesting here.

**Reasons To Reject:**

* It covers limited number of tasks here. The paper has proposed its own benchmarks. It will be interesting to see how the method applies to broader domains.

* The method is complex. The paper did not show limitations or error analysis for when the system would fail.

---

> ### Author Response · Authors · 2025-06-02
>
> We thank Reviewer Y93Q for recognizing the practical impact of our cost savings and the novelty of our proposed framework.
>
> ---
>
> R1: `It covers limited number of tasks here. The paper has proposed its own benchmarks. It will be interesting to see how the method applies to broader domains.`
>
> We agree that broader applicability is important. However, the challenge is that our focus is on a **streaming setup**, where an agent is deployed and then asked to solve a stream of related problem instances in a cost-effective manner. This is realistic---it corresponds to the case where an agent is synthesized and then deployed as a service---but it does not match existing benchmarks.  Current agentic and tool-use datasets (e.g., APIBench, ToolBench, APIBank, GAIA, M&Ms, GTA [1-6]) only provide **one input per agentic program**, which is the reason we needed to create our own benchmark tasks.
>
> That said, our framework is **task-agnostic**. The policy operates over abstract function call graphs and adapts to empirical cost/accuracy profiles, making it directly applicable to any domain—vision, text, speech, or multimodal—so long as the task can be expressed as an FMP. In all such cases, our no-regret guarantee (Appendix A) and optimization routine apply without modification. We will clarify this generality in the final version.
>
> ---
>
> R2: `The method is complex. The paper did not show limitations or error analysis for when the system would fail.`
>
> Thank you for this valuable feedback. To address your concern about the complexity of our method, we have added an ablation study to compare our structured policy with a more straightforward, full-policy method. The results on (a) the Streaming Binary VQA benchmark (https://ibb.co/XfWVzhVV) and (b) the Streaming Open-form VQA benchmark (https://ibb.co/F4fvTBnF) validate that the full-policy method (yellow line) significantly underperforms, which justifies our current design.
>
> While the core algorithm is modular and built from well-understood components (e.g., Thompson sampling and REINFORCE), we acknowledge that our system’s performance depends on several factors. We will add a discussion of the following limitations in the final version:
>
> - **Incorrect workflow structure**: If the LLM-generated FMP does not match the true task logic, the routing policy may optimize an invalid execution path. However, we find that modern LLMs (e.g., GPT-4o) produce robust FMPs, and their structured form makes inconsistencies easy to audit and correct. Moreover, our core contribution — the dynamic online resource allocation framework — fully supports human-written FMPs, which are common in real-world deployments.
> - **Incapable neural backends**: If the available backends are too weak for a sub-task (e.g., answering a fine-grained VQA with only small backends), the policy may still fail despite optimal routing. This is illustrated in Appendix Fig. 7, where using less capable models leads to noticeable drops in accuracy and F1. However, this reflects a limitation of the backend pool, not of the framework itself.
> ---
> Reference:
>
> [1] Patil, Shishir G., et al. "Gorilla: Large language model connected with massive apis." Advances in Neural Information Processing Systems 37 (2024): 126544-126565.
>
> [2] Qin, Yujia, et al. "Toolllm: Facilitating large language models to master 16000+ real-world apis." arXiv preprint arXiv:2307.16789 (2023).
>
> [3] Li, Minghao, et al. "Api-bank: A comprehensive benchmark for tool-augmented llms." arXiv preprint arXiv:2304.08244 (2023).
>
> [4] Mialon, Grégoire, et al. "Gaia: a benchmark for general ai assistants." The Twelfth International Conference on Learning Representations. 2023.
>
> [5] Ma, Zixian, et al. "m & m’s: A Benchmark to Evaluate Tool-Use for m ulti-step m ulti-modal Tasks." European Conference on Computer Vision. Cham: Springer Nature Switzerland, 2024.
>
> [6] Wang, Jize, et al. "GTA: a benchmark for general tool agents." The Thirty-eight Conference on Neural Information Processing Systems Datasets and Benchmarks Track. 2024.

---

> > ### Comment · Reviewer_Y93Q · 2025-06-05
> >
> > Thank you for the response. I don't have further questions.

---

> > > ### Author Response · Authors · 2025-06-09
> > >
> > > Thank you for your follow-up and for engaging with our rebuttal. If any additional questions arise, we’d be glad to address them during the discussion period. If our responses and the supplementary results have satisfactorily addressed your concerns, we would be grateful if you would consider reflecting that in your final assessment. Thank you again for your time!

---

### Decision · Program_Chairs · 2025-07-08

**Decision:**

Accept

**Comment:**

This paper introduces foundation model programs (FMPs), offering a solution to the growing challenge of inference-time resource costs for large foundation models. By translating tasks into programs and learning policies for resource allocation, the proposed method allows simpler tasks to be handled by smaller, cheaper models, while more complex tasks utilize larger, more capable models. The authors also introduce two benchmarks for binary and open-form visual question answering. The proposed approach demonstrates significant resource savings with minimal accuracy loss.

Pros:
- The proposed approach is suitable for real-world deployment where computational resources are constrained.
- The proposed streaming tasks are new for agentic evaluation.
- The authors conduct comprehensive analyses of the proposed FMPs and new benchmarks.

Cons:
- The implementation of the system, which includes code generation, program synthesis, and dynamic backend selection, is complex and may be challenging to integrate into existing systems. The authors promise to add the discussion of the incorrect workflow structure or incapable neural backends in the final version.
- Limited Evaluation: The manuscript only reports their performance on streaming tasks. The authors claim the proposed approach is modality-agnostic and promise to add more discussion in the final version.